# Adversarially Robust Spiking Neural Networks with Sparse Connectivity

Mathias Schmolli[1], Maximilian Baronig[1,2], Robert Legenstein[1], Ozan Özdenizci[1,3]

[1] Institute of Machine Learning and Neural Computation, Graz University of Technology, Austria
[2] TU Graz - SAL Dependable Embedded Systems Lab, Silicon Austria Labs, Austria
[3] Chair of Cyber-Physical-Systems, Montanuniversität Leoben, Austria

m.schmolli@alumni.tugraz.at,{baronig,robert.legenstein,oezdenizci}@tugraz.at

Deployment of deep neural networks in resource-constrained embedded systems requires innovative algorithmic solutions to facilitate their energy and memory efficiency. To further ensure the reliability of these systems against malicious actors, recent works have extensively studied adversarial robustness of existing architectures. Our work focuses on the intersection of adversarial robustness, memory- and energy-efficiency in neural networks. We introduce a neural network conversion algorithm designed to produce sparse and adversarially robust spiking neural networks (SNNs) by leveraging the sparse connectivity and weights from a robustly pretrained artificial neural network (ANN). Our approach combines the energy-efficient architecture of SNNs with a novel conversion algorithm, leading to state-of-the-art performance with enhanced energy and memory efficiency through sparse connectivity and activations. Our models are shown to achieve up to $100\times$ reduction in the number of weights to be stored in memory, with an estimated $8.6\times$ increase in energy efficiency compared to dense SNNs, while maintaining high performance and robustness against adversarial threats.

## 1. Introduction

Resource-efficient deep learning often requires novel algorithmic solutions that can jointly address various considerations of a machine learning system, such as energy efficiency, memory usage, or security and reliability. Considering the latter, recent works have extensively studied adversarial training based algorithms with traditional artificial neural network (ANN) architectures, to improve robustness against adversarial attacks [1]. In the context of adversarially robust optimization, achieving memory efficiency through sparsity in the utilized ANN architecture has been shown to be a challenging problem, since robustness typically improves with increasing network complexity [2, 3]. Consequently, several studies have explored robustness aware pruning methods for adversarially pretrained dense neural networks [4–6].

Following another line of work, there has been increasing interest in achieving adversarial robustness with spiking neural network (SNN) architectures [7–11]. The primary motivation of this research problem is the energy-efficiency potential of these models in reliable edge computing applications, since SNNs operate using binary signals that are transmitted over time, which reduces the need for computationally expensive matrix multiplications found in traditional ANNs, and yields energy-efficient event-driven processing capabilities [12, 13]. Along this direction, our work focuses on the intersection of achieving *adversarial robustness* with inherently *energy-efficient* SNN architectures using *sparse connectivity* structures, which has not been previously studied.

Our initial empirical analyses reveal that achieving robustness through standard end-to-end adversarial training with highly sparse feedforward SNNs is computationally challenging and infeasible with spike-based backpropagation through time (BPTT) [14, 15]. Accordingly, we tackle this problem by exploiting stable adversarial training capabilities of ANNs. We introduce a neural network conversion algorithm that leverages the sparse connectivity structure and weights of a robustly pretrained and pruned ANN, to obtain an adversarially robust and sparse SNN. Our method can

Second Conference on Parsimony and Learning (CPAL 2025).

successfully generate feedforward SNNs with layerwise uniform or non-uniform sparse connectivity structures, by utilizing existing adversarial training and robustness aware pruning methods designed for ANNs. Contributions of this work are summarized as follows:

- We present a robust and sparse ANN-to-SNN conversion algorithm, combined with a post-conversion sparse SNN finetuning phase. Our method can inherit robustness properties of an adversarially pretrained ANN to the resulting SNN under strict sparsity constraints.

- Our algorithm offers the first solution to the problem of achieving sparsity with adversarially robust feedforward SNNs, and scales to larger benchmarks such as TinyImageNet classification, which is not commonly explored in previous SNN robustness studies. Our models are shown to achieve up to $100\times$ compression rates, with an estimated $8.6\times$ increase in energy-efficiency over their dense SNN counterparts.

- Our approach is agnostic to the adversarial ANN pretraining and robust ANN pruning methods, thus allows the integration of existing and future advancements in adversarially robust training methods in the ANN domain.

## 2. Background

### 2.1. Spiking Neural Networks

Spiking neural networks perform event-based information processing with binary activations [16]. Leaky-integrate-and-fire (LIF) neurons in feed-forward SNNs operate according to the dynamics:

$$\boldsymbol{v}^l(t^-) = \tau \boldsymbol{v}^l(t-1) + \boldsymbol{W}^l \boldsymbol{o}^{l-1}(t), \tag{1}$$

$$\boldsymbol{o}^l(t) = H(\boldsymbol{v}^l(t^-) - V_{th}^l), \tag{2}$$

$$\boldsymbol{v}^l(t) = \boldsymbol{v}^l(t^-)(1 - \boldsymbol{o}^l(t)), \tag{3}$$

where the output spikes from the previous layer $\boldsymbol{o}^{l-1}(t)$ are weighted by the synaptic connectivity matrix $\boldsymbol{W}^l$, the membrane potential leak factor is denoted by $\tau$, the neuron membrane potential before and after firing of a spike at time $t$ is denoted by $\boldsymbol{v}^l(t^-)$ and $\boldsymbol{v}^l(t)$, $H(.)$ denotes the Heaviside step function, and the neuron firing threshold is denoted by $V_{th}^l$. For LIF neurons with a hard-reset mechanism, if the neuron emits a spike at time step $t$, the membrane potential is reset to zero via Eq. (3). We provide the input $\boldsymbol{o}^0(t)$ to the first layer via *direct coding*, i.e., the input signal intensity (e.g., image pixel) values are applied to the first layer neurons for $T$ simulation timesteps.

Gradient based SNN training can be accomplished via backpropagation through time (BPTT) [14, 15, 17], which requires the use of surrogate gradient functions to approximate the discontinuous derivative of the spike function [18]. Another line of methods obtain SNNs via ANN-to-SNN conversion [19]. This is achieved by harnessing pretrained ANN weights and tuning the firing rates of spiking neurons in SNNs to be approximately proportional to the pretrained ANN neuron activations [20]. Although earlier conversion methods required high inference latencies [21], recent *hybrid conversion* methods alleviated this by using post-conversion BPTT-based finetuning, such that good performance with short inference latencies (<10 timesteps) can be achieved [22, 23].

**Energy-Efficiency of SNNs:** Deep ANNs often require large weight matrix multiplications with floating point multiplication and accumulation (MAC) operations. In SNNs, however, layer activations are binary, thus only accumulation operations (AC) are performed at the synaptic connections in an event-driven manner, i.e., only when a spike is emitted from pre-synaptic neurons. Overall energy consumption of MAC operations are significantly higher than AC operations on various types of hardware [24]. Taking this into account, feedforward SNNs can result in an advantage of energy-efficiency if the spiking activity across the simulation duration remains sufficiently low [12, 13].

### 2.2. Adversarial Robustness of Neural Networks

Neural networks are vulnerable against *adversarial examples* with imperceptible test-time perturbations that lead to incorrect decisions [1]. Prominent white-box threats such as fast gradient

sign method (FGSM) [25] and projected gradient descent (PGD) [2] generate $l_\infty$-norm bounded perturbations based on the gradient, where $\tilde{\boldsymbol{x}} = \boldsymbol{x} + \epsilon \cdot \text{sign}(\nabla_{\boldsymbol{x}}\mathcal{L}(f(\boldsymbol{x};\boldsymbol{\theta}),y)))$ indicates the FGSM attack with an $\epsilon > 0$ perturbation strength, and PGD is an iterative variant: $\tilde{\boldsymbol{x}}_{k+1} = \prod_\epsilon^\infty [\tilde{\boldsymbol{x}}_k + \alpha \cdot \text{sign}(\nabla_{\tilde{\boldsymbol{x}}_k}\mathcal{L}(f(\tilde{\boldsymbol{x}}_k;\boldsymbol{\theta}),y))]$, with $\prod_\epsilon^\infty$ being the clipping onto the $l_\infty$-norm $\epsilon$-ball around $\boldsymbol{x}$, and $\alpha$ is the perturbation step size. To counter these attacks, most effective empirical defenses rely on *adversarial training (AT)*, by incorporating adversarial examples into the training process:

$$\min_{\boldsymbol{\theta}} \mathbb{E}_{(\boldsymbol{x},y)\sim\mathcal{D}} \left[ \max_{\tilde{\boldsymbol{x}}\in\mathcal{B}_\epsilon^\infty(\boldsymbol{x})} \mathcal{L}_{\text{robust}}(f(\tilde{\boldsymbol{x}};\boldsymbol{\theta}),y) \right], \tag{4}$$

with the inner maximization obtaining adversarial samples in $\mathcal{B}_\epsilon^\infty(\boldsymbol{x}) := \{\tilde{\boldsymbol{x}} : \|\tilde{\boldsymbol{x}} - \boldsymbol{x}\|_\infty \leq \epsilon\}$. Standard AT [2] only uses adversarial samples via $\mathcal{L}_{\text{robust}}^{\text{AT}} = -\log p(y|\tilde{\boldsymbol{x}};\boldsymbol{\theta})$. Later methods such as TRADES [26], MART [27], or consistency regularization [28] introduced unique regularization methods to achieve better robustness-accuracy trade-offs.

**Adversarial Robustness of SNNs:** Earlier works aimed to improve adversarial robustness of SNNs based on methods designed for ANNs [29, 30] (e.g., standard AT with BPTT [31, 32]). Subsequently, various studies examined structural SNN components and their contributions to robustness [7, 33, 34]. Majority of recent methods to improve robustness in SNNs rely on regularized end-to-end AT approaches with BPTT [8–10, 35], which requires computationally heavy optimization procedures for large-scale problems. Accordingly, from a different perspective, hybrid ANN-to-SNN conversion with adversarially pretrained robust ANNs has achieved state-of-the-art adversarial robustness in SNNs [11], alleviating the computational burden of end-to-end AT with BPTT.

## 2.3. Sparsity and Adversarial Robustness

Neural networks often need to meet certain resource constraints in terms of their memory usage on embedded systems. This has been widely studied in the context of achieving sparsity in models to reduce the number of weights that need to be stored in memory [36]. Accordingly, previous works have also explored sparsity in adversarially trained models via *robust pruning* algorithms [5, 6].

**Neural Network Pruning:** Given a model $f(\boldsymbol{x};\boldsymbol{\theta})$, a pruning algorithm aims to find a global pruning mask $\boldsymbol{m}$ consisting of per-layer binary matrices $\{\boldsymbol{M}^{(l)}\}_{l=1}^L$, which flags a desired global sparsity ratio $\kappa \in [0,1)$ of parameters $\boldsymbol{\theta}$ from the dense model that will be set to zero. The goal is to obtain a well-performing new model $f(\boldsymbol{x};\boldsymbol{m}\odot\boldsymbol{\theta}')$ with $\|\boldsymbol{m}\odot\boldsymbol{\theta}'\|_0 \leq (1-\kappa)\cdot n$, where $n$ is the total number of parameters and $\kappa \in [0,1)$ (e.g., $\kappa = 0.99$ indicates 99% sparsity, $\kappa = 0$ indicates the dense model). After determining $\boldsymbol{m}$, remaining parameters are finetuned to recover performance, such that $\boldsymbol{\theta}'$ deviates from the original $\boldsymbol{\theta}$. During pruning, the significance of each parameter is generally quantified via an importance score vector $\boldsymbol{s}$, which determines the mask indices $m_j$ based on $s_j$:

$$m_j = \mathbb{1}[s_j - \hat{s}_\sigma \geq 0], \quad \forall j \in \{1,\ldots,n\}, \tag{5}$$

where $\sigma = (1-\kappa)\cdot n$, $\hat{\mathbf{s}} = \text{SortDescending}(\mathbf{s})$, thus $\hat{s}_\sigma$ is the $\sigma$-th largest element in $\mathbf{s}$, and $\mathbb{1}[.]$ is the indicator function. A baseline approach is least weight magnitude (LWM) based pruning [36]:

$$s_j = |\theta_j|, \quad \forall j \in \{1,\ldots,n\}. \tag{6}$$

Here, the pruning rate $\kappa$ is generally applied per-layer to the model such that layerwise uniform sparsity is maintained. Importantly, LWM can not consider any adversarial robustness criterion while choosing which parameters to prune.

**Robust Pruning with Learned Importance Scores:** One of the earlier robustness-aware pruning methods, HYDRA [5], optimizes $\boldsymbol{s}$ as a learnable parameter under an AT objective. Specifically, the importance scores are updated with gradient descent (without updating model weights) via:

$$\min_{\mathbf{s}} \mathbb{E}_{(\boldsymbol{x},y)\sim\mathcal{D}} \left[ \max_{\tilde{\boldsymbol{x}}\in\mathcal{B}_\epsilon^\infty(\boldsymbol{x})} \mathcal{L}_{\text{robust}}(f(\tilde{\boldsymbol{x}};\mathbf{m}(\mathbf{s})\odot\boldsymbol{\theta}),y) \right], \tag{7}$$

where in each forward pass the current mask $\mathbf{m}(\mathbf{s})$ is obtained via Eq. (5) by maintaining *uniform per-layer sparsity* (i.e., highest ranked $\sigma^{(l)} = (1-\kappa)\cdot n^{(l)}$ scores in the $l$-th layer remain). All scores $s_j$ are optimized by using a straight-through estimator through the mask during backpropagation [37].

The recent holistic adversarially robust pruning (HARP) method [6] advanced this idea by simultaneously optimizing how many and which parameters to prune individually in each layer. This yields *non-uniform per-layer sparsity* ratios that differ from the global sparsity rate $\kappa$. This is performed by introducing a learnable, continuous-valued representation of layerwise compression rates, such that Eq. (7) is modified to also optimize layerwise compression quotas $r^{(l)}$ alongside $s$, together with an additional regularizer term that enforces the global sparsity constraint of $\kappa$ based on $r^{(l)}$.

Finally, the resulting mask $\mathbf{m}$ based on the optimized $\mathbf{s}^\star$ is used for final pruning and subsequent adversarial finetuning, yielding an adversarially robust and sparse model with parameters $\mathbf{m} \odot \boldsymbol{\theta}'$.

# 3. Adversarially Robust and Sparse ANN-to-SNN Conversion

Our approach requires a baseline ANN for conversion, hence we first perform dense ANN adversarial pretraining and pruning. We then leverage this sparse connectivity and weights via ANN-to-SNN conversion. Finally, we adversarially finetune the sparse SNN to improve performance.

## 3.1. Pretraining Robust and Sparse ANNs

The initial stage of our method can exploit any existing AT algorithm designed for ANNs, following the objective in Eq. (4) for any chosen adversarial perturbation budget $\epsilon$. After dense ANN adversarial pretraining, we perform robust pruning with learned importance scores via Eq. (7) to determine which connections to prune from the dense ANN for a given global sparsity ratio $\kappa$. This is achieved either by preserving *layerwise uniform sparsity* [5], or by learning layer compression rates (i.e., *layerwise non-uniform sparsity* [6]), such that we obtain per-layer sparsity masks $\{\boldsymbol{M}^{(l)}\}_{l=1}^{L}$.

## 3.2. Converting Pretrained ANNs to SNNs

Our conversion approach relies on the conventional threshold-balancing algorithm [21, 23], which was also recently explored for robust ANN-to-SNN conversion with *dense* connectivity [11].

**Parameter Initialization from Pretrained ANN:** We use the same network architecture for the SNN using the weights $\{\boldsymbol{W}^{(l)}\}_{l=1}^{L}$ from the pretrained, sparse and robust ANN as the synaptic connectivity weights, considering the per-layer sparsities imposed via $\{\boldsymbol{M}^{(l)}\}_{l=1}^{L}$. We exchange ReLU activations of the ANN with LIF spiking neuron dynamics. We use threshold-dependent batch-norm operations in the SNN [38] to replace ANN batch-norm layers, where weighted spiking inputs are temporally gathered in $\bar{\boldsymbol{O}}^l = [\boldsymbol{W}^l \boldsymbol{o}^{l-1}(1), \dots, \boldsymbol{W}^l \boldsymbol{o}^{l-1}(T)]$, and batch-norm is performed on $\bar{\boldsymbol{O}}^l$, i.e., mean and variance statistics are estimated over both mini-batches and time. We then re-use the pretrained ANN batch-norm affine transformation parameters $\varphi^l$ and $\omega^l$ for the SNN, but now estimate the running average statistics from scratch during SNN finetuning based on the spiking data [11].

**Initialization of Firing Thresholds:** Following weight initialization, we calibrate per-layer firing thresholds of the SNN by using a number of training mini-batch forward passes. Specifically, we generate inputs to the network via direct coding for a number of calibration timesteps $T_c > T$. Subsequently, starting from the first layer, we observe the pre-activation values (sum of the weighted spiking inputs) across $T_c$ timesteps, and set the firing threshold for neurons in that layer to be the $\rho$-th percentile value of the distribution of pre-activations, as performed in [39]. We then set the firing thresholds for all other layers in the same way, and thus estimate proportionally balanced firing threshold values as in [21, 23]. Finally, we scale our initial estimates with a factor $\lambda < 1$ in order to promote a better spike flow early on during finetuning, and make these firing thresholds $\{V_{th}^l\}_{l=1}^{L-1}$ *trainable* during the finetuning process (see Appendix A.2 for details on hyperparameters).

## 3.3. Adversarial Finetuning of Sparse SNNs

We perform post-conversion robust finetuning of our sparse SNNs, similar to hybrid ANN-to-SNN conversion methods. This is mainly necessary to calibrate the restructured spatio-temporal SNN batch-norm layers, and to adversarially finetune the weights and firing thresholds for a desired $T$.

We denote the model predictions for an adversarial and a clean example as $f(\tilde{\boldsymbol{x}}; \mathbf{m} \odot \boldsymbol{\theta})$ and $f(\boldsymbol{x}; \mathbf{m} \odot \boldsymbol{\theta})$. These values are estimated by the sparse SNN via the integrated non-leaky membrane potential of output neurons over $T$ timesteps. Accordingly, we use a regularized robust finetuning objective that facilitates stability of the output neuron membrane potentials under adversarial attacks [26]:

$$\min_{\boldsymbol{\theta}} \mathbb{E} \left[ \mathcal{L}\left(f(\boldsymbol{x}; \mathbf{m} \odot \boldsymbol{\theta}), y\right) + \beta \cdot \max_{\tilde{\boldsymbol{x}} \in \mathcal{B}_\epsilon^\infty(\boldsymbol{x})} D_{\mathrm{KL}}\left(f(\tilde{\boldsymbol{x}}; \mathbf{m} \odot \boldsymbol{\theta}) \| f(\boldsymbol{x}; \mathbf{m} \odot \boldsymbol{\theta})\right) \right], \tag{8}$$

such that $\beta$ controls the robustness-accuracy trade-off during finetuning. Specifically, we adversarially finetune parameters $\boldsymbol{\theta}$ which consist of $\{\boldsymbol{W}^l, \varphi^l, \omega^l\}_{l=1}^L$ and $\{V_{th}^l\}_{l=1}^{L-1}$. The computational load of finetuning is mainly determined by the method used for inner maximization to obtain $\tilde{\boldsymbol{x}}$, which can be performed efficiently via single-step attacks as conventionally done in SNN studies [8, 35].

Our finetuning objective is optimized via spike-based BPTT, which necessitates surrogate gradients to backpropagate the error through the non-differentiable function in Eq. (2). We use piecewise linear surrogate gradients via: $\frac{\partial o^l(t)}{\partial v^l(t)} = \frac{1}{\gamma_w^2} \cdot \max\left\{0, \gamma_w - |v^l(t^-) - V_{th}^l|\right\}$, with $\gamma_w = 1$ during training. For attack evaluations, we consider adaptive SNN adversaries that can vary both $\gamma_w$ and the shape of the surrogate function to reliably estimate adversarial perturbations.

**Maintaining Sparsity During Finetuning:** We preserve the transferred sparse ANN connectivity during the finetuning phase and keep the network connectivity frozen. This is performed by applying the global pruning mask $\boldsymbol{m}$, consisting of per-layer binary masks $\{\boldsymbol{M}^l\}_{l=1}^L$ that were obtained during the robust ANN pruning phase, at each forward pass operation and weight update:

$$\boldsymbol{W}^l \leftarrow \boldsymbol{W}^l - \eta\left(\Delta \boldsymbol{W}^l \odot \boldsymbol{M}^l\right). \tag{9}$$

Our resulting SNN therefore maintains the same sparsity as the pretrained ANN, but with new parameters $\boldsymbol{\theta}'$ which are adversarially finetuned via spike-based BPTT.

# 4. Experiments

## 4.1. Experimental Setup

**Datasets and Models:** We experimented with CIFAR-10/100 and TinyImageNet datasets, using VGG-11, VGG-16 and WideResNet architectures with depth 28 and widening factor of 4, i.e., WRN-28-4. All SNNs considered in this study were run for $T = 8$ timesteps with IF neurons ($\tau = 1$).

**Adversarial Pretraining of Dense ANNs:** We trained baseline ANNs via the TRADES loss [26] for 100 epochs, using 10-step PGD with $\epsilon = 2/255$ to craft adversarial examples in the inner maximization stage. For dense ANNs pretrained on CIFAR-10, additional pseudo-labeled data samples during AT were also utilized, which was shown to improve robustness [40].

**Pruning Adversarially Robust ANNs:** We optimize robust weight importance scores for 20 epochs based on the configurations from HYDRA (for layerwise *uniformly* sparse) and HARP (for layerwise *non-uniformly* sparse) models. We also perform naive LWM based pruning as a baseline. We consider all dense and convolutional layer weights subject to unstructured pruning. After pruning the ANNs to a desired sparsity ratio $\kappa$, we finetune sparse ANNs for 30 epochs following [5, 6].

**Adversarial Finetuning of Sparse SNNs:** We perform adversarial SNN finetuning following Eq (8) with BPTT for 80 epochs, using random-step FGSM (RFGSM) [41] with $\epsilon = 2/255$ in the inner maximization stage. We set $\beta = 2$ in all experiments, unless specified otherwise. During conversion, we initialized per-layer firing thresholds with the $\rho = 99.7\%$ percentile of the pre-activation values that we observed over $T_c = 100$ calibration timesteps, and use $\lambda = 0.3$ as the scaling factor.

**Baseline Comparisons to End-to-End Sparse AT:** As there are no existing solutions to this problem, we compare our conversion-based approach with end-to-end adversarial SNN training with a static sparse connectivity, using the same objective in Eq. (8). This connectivity was determined either randomly by uniform sampling, or set as the same sparse connectivity from the pretrained sparse ANN. These networks were adversarially trained from scratch for at most 350 epochs.

## 4.2. Evaluating Adversarial Robustness of SNNs

The success of gradient-based attacks on SNNs strictly depend on the surrogate gradient used for BPTT [34]. The naive common practice is to use the same surrogate gradient for the attack as the one used during training [8, 9], which might lead to ineffective attacks and give a false sense of security for SNNs. Accordingly, recent works have assessed SNN robustness using an *ensemble* attack benchmark, where the adversary does not only utilize a single gradient approximation path, but via an ensemble of various surrogate gradient options during BPTT until successful [10, 11] (see Appendix A.3 for details). This significantly increases attack effectiveness against SNNs and yields a worst case evaluation scenario [42, 43] (see Table A1 for an ensemble attack analysis).

Accordingly, we evaluate SNNs against *ensemble* attacks with FGSM [25], PGD [2], Auto-PGD with difference of logits ratio (APGD-DLR), and with cross-entropy loss (APGD-CE) [44]. Main PGD attacks were run with 10 steps (independently for each surrogate gradient in the ensemble), with step size $\alpha = 2.5 \times \epsilon /\#$ steps. In TinyImageNet experiments, we also evaluated SNNs against rate gradient approximation (RGA) attacks combined with 10-step PGD, which are designed for SNNs [45]. We evaluated black-box robustness with Square Attack [46], using various numbers of limited queries.

## 4.3. Estimation of Energy Consumption in SNNs

Energy-efficiency in feedforward SNNs is tightly coupled with the overall spiking activity during inference. Besides the memory-efficiency offered by sparse connectivities, we assess the impact of sparsity on the model's spiking activity, to verify that targeting sparsity does not degrade the energy-efficiency of the robust SNN as opposed to its dense counterpart due to higher spike rates.

Following the notation from [47], we denote the number of *active* outgoing connections from the $i$-th neuron in layer $l$ by $\psi_{l,i}$, i.e., the number of non-zero elements in the matrix $(\boldsymbol{W}^{l+1})_{*,i}$. For densely connected feedforward SNNs with $\kappa = 0$ this would result in:

$$\psi_{l,i} = \psi_l = c_{l+1} \qquad \text{(linear layer)}, \qquad (10)$$

$$\psi_{l,i} \simeq \psi_l = \frac{w_{l+1}h_{l+1}}{w_l h_l} c_{l+1} k_{l+1}^2 \qquad \text{(conv layer)}, \qquad (11)$$

where $c_l$ is the number of filters, $w_l, h_l$ are the feature map dimensions, and $k_l$ is the kernel width. We use $\frac{w_{l+1}h_{l+1}}{w_l h_l}$ for normalization if pooling is performed between layers. For sparse SNNs, we specifically take into account the exact connectivity structure and calculate all outgoing connections for every neuron to obtain $\psi_{l,i}$. This yields an accurate estimate since only spikes over non-zero synaptic weights result in AC operations. Subsequently, we estimate the energy consumption via:

$$E_{\text{SNN}} := \sum_{l=1}^{L} T \cdot E_{\text{AC}} \cdot \mathop{\mathbb{E}}_{\boldsymbol{x} \in \mathcal{X}} \left[ \sum_{i=1}^{d_l} \psi_{l,i} o_{l,i} \right], \qquad (12)$$

where $E_{\text{AC}}$ denotes the energy consumption of a single accumulate operation, $d_l$ is the number of neurons in the $l$-th layer, $o_{l,i} \in \{0, 1\}$ are the output spikes emitted by neuron $i$ in the $l$-th layer, and $\mathbb{E}_{\boldsymbol{x} \in \mathcal{X}}[.]$ indicates taking the empirical expectation over the available test dataset $\mathcal{X}$.

It is important to note that we will use Eq. (12) to estimate *relative* energy-efficiency gains of achieving sparsity in the SNNs, in relation to their dense SNN counterparts, in the context of preserving adversarial robustness under sparsity constraints. We do not intend to compare SNNs with ANNs, since this would require some heuristic assumption on the relationship between $E_{\text{AC}}$ and $E_{\text{MAC}}$ [24].

# 5. Experimental Results

## 5.1. Robust ANN-to-SNN Conversion Outperforms End-to-End Sparse AT

We compare our approach with end-to-end AT of sparse SNNs, to empirically demonstrate the necessity of a novel ANN-to-SNN conversion-based solution, to the problem of achieving sparsity in robust SNNs. Figure 1 shows that for 90% sparse SNNs (layerwise uniform sparsity), our

Table 1: Detailed evaluations of 90% sparse VGG-16 models with layerwise uniform sparsity on CIFAR-10. We used white-box attacks with $\epsilon = 2/255 \,/\, 4/255 \,/\, 8/255$, and black-box SquareAttack with $\epsilon = 8/255$ using 500 / 1000 / 5000 queries. For SNNs, white-box attacks are implemented using a surrogate gradient ensemble. SquareAttack was run identically for SNNs and ANNs.

| | Sparse & Robust SNN | | | | Sparse & Robust ANN |
| | End-to-End Adv. Training | | Conversion + Sparse FT (Ours) | | |
| | Random Conn. | ANN Conn. | | | |
| --- | --- | --- | --- | --- | --- |
| Clean Acc. | 87.4 | 88.2 | **89.5** | Clean Acc. | 92.3 |
| $\text{FGSM}_{ens}$ | 65.6 / 50.8 / 27.8 | 64.9 / 49.8 / 26.5 | **71.6 / 61.0 / 41.1** | FGSM | 81.3 / 67.4 / 42.5 |
| $\text{PGD}_{ens}$ | 62.4 / 42.2 / 11.9 | 61.1 / 40.1 / 9.70 | **69.9 / 54.7 / 26.1** | PGD | 79.8 / 59.5 / 22.2 |
| $\text{APGD-DLR}_{ens}$ | 58.3 / 40.8 / 12.8 | 57.9 / 38.8 / 10.4 | **65.2 / 52.0 / 26.9** | APGD-DLR | 79.4 / 58.5 / 21.2 |
| $\text{APGD-CE}_{ens}$ | 55.7 / 36.5 / 8.80 | 54.7 / 34.2 / 7.10 | **62.6 / 47.9 / 21.1** | APGD-CE | 79.5 / 57.9 / 19.3 |
| SquareAttack | 47.9 / 42.4 / 32.2 | 49.8 / 43.8 / 33.9 | **52.3 / 46.5 / 36.3** | SquareAttack | 61.1 / 51.6 / 32.7 |

method outperforms both end-to-end AT models throughout training (i.e., clean/robust acc. at the end of training, E2E w/random conn.: 87.4/11.9, E2E w/ANN conn.: 88.2/9.7, Ours: 89.5/26.1). Our results highlight the importance of the robust ANN based weight initialization alongside the sparse connectivity structure. Importantly, we achieve this result with a more computationally efficient method (i.e., total wall-clock time: E2E SNN AT: 40.6 hours vs Ours: 24.5 hours, see Appendix A.2 for details). This is due to the majority of our robust optimization process being performed in the ANN domain, without the computational overhead of AT with BPTT.

We demonstrate in Table 1 that our conversion method is superior in terms of robustness across different attacks. We obtained sparse SNNs that are more than twice as robust against strong adversaries, than E2E adversarially trained models (e.g., robust acc. under $\text{APGD-CE}_{ens}$ with $\epsilon = 8/255$, E2E w/random conn.: 8.8%, E2E w/ANN conn.: 7.1%, Ours: 21.1%). Note that we tried adapting existing end-to-end regularized AT based SNN robustness

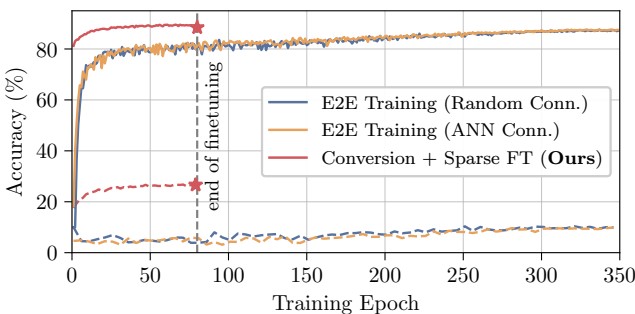

Figure 1: Comparing our method with end-to-end (E2E) AT, via clean (solid) and robust (dashed) accuracies evaluated with $\text{PGD}_{ens}$ at $\epsilon = 8/255$ (90% sparse VGG-16 SNN on CIFAR-10). Sparsity in E2E training was initialized either randomly, or by using the ANN connectivity as in our method.

methods [8, 10] to improve performance of E2E trained models with 90% sparsity. However, we observed that such regularization schemes made the highly sparse training process sensitive to hyperparameters and unstable, and we were not successful in adapting these methods to this problem.

**Robustness Against Black-box Attacks:** Evaluations with white-box attacks on SNNs and ANNs are not necessarily identical or directly comparable, since SNN adversaries require a surrogate gradient ensemble approach. However, Square Attack presents a fairer comparison since these query-based attacks are run identically without the use of gradients. We observe that upon conversion our sparse SNN shows higher resilience after a high volume of available queries (robust acc. at 5000 queries, Our SNN: 36.3%, ANN: 32.7%). For completeness, we also compared the 90% sparse ANN and converted SNN with layerwise *non-uniform* sparsity. Our SNNs yielded stronger robustness than its ANN counterpart also in this setting (at 5000 queries, our SNN: 42.2%, ANN: 37.0%).

## 5.2. Adversarial Robustness of Sparse SNNs

Figure 2 presents detailed evaluations of our method with both layerwise uniform and non-uniform robust ANN pruning strategies, at various compression rates. We also indicate a dense ANN-to-

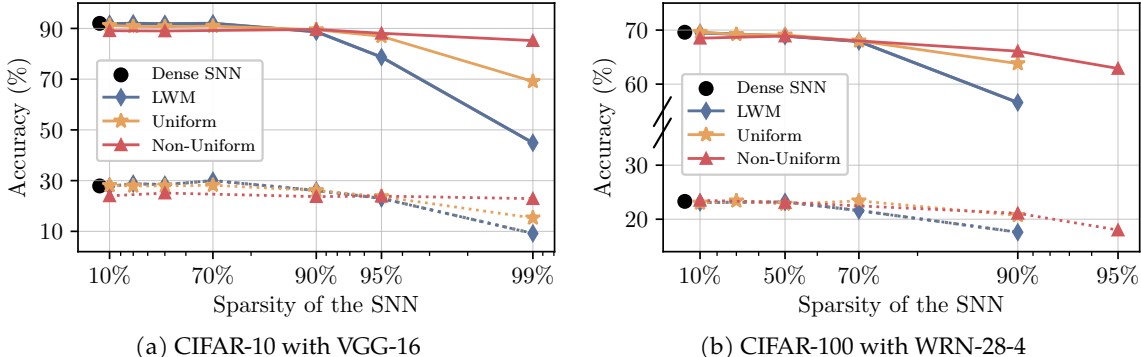

|  | (a) CIFAR-10 with VGG-16 | (b) CIFAR-100 with WRN-28-4 |

Figure 2: Clean (solid lines) and robust (dashed lines) accuracies of converted SNNs evaluated with PGD$_{ens}$ at $\epsilon = 8/255$ for CIFAR-10 and $\epsilon = 4/255$ or CIFAR-100, for different sparsity levels. ANNs used for conversion were either dense, pruned with LWM, or layerwise uniformly or non-uniformly sparse through learned importance scores (see Appendix B.1 for numerical details of all results).

SNN conversion baseline, and LWM as a naive ANN pruning baseline. Overall, we empirically validate that our approach generalizes well across different robust ANN pruning strategies, and we can reach any desired compression rate with an anticipated trade-off in clean and robust accuracies. Notably, in our CIFAR-10 experiments, SNN robustness was not impacted up to 70% sparsity (i.e., clean/robust acc., Dense SNN: 92.0/27.9, Uniform (70%): 90.9/28.2). We observed that layerwise non-uniform sparsity (shown with red lines) helps in maintaining robustness at high compression rates. Specifically, we achieved clean/robust acc. under PGD$_{ens}$ for CIFAR-10 with 99% sparsity (100$\times$ compression), LWM: 44.9/9.2, Uniform: 69.1/15.3, Non-Uniform: 85.2/22.9, and for CIFAR-100 with 90% sparsity (10$\times$ compression), LWM: 56.6/17.6, Uniform: 63.8/20.7, Non-Uniform: 66.1/21.1 (see Table B3 for PGD$_{ens}$ attack evaluations under various number of iterations).

In Appendix B.2, we investigate the influence of ANN pretraining on the robustness of the converted SNN. We observed that more robust ANN weights result in better SNNs. Specifically, we obtained 90% sparse SNNs with clean/robust acc. of 83.2/40.0, when we performed heavier adversarial pretraining of the ANN using $\epsilon = 8/255$, as opposed to the 89.5/26.1 from Table 1.

## 5.3. Experiments on TinyImageNet

We validate our method on the larger scale TinyImageNet dataset, which is generally not explored in the field of robust SNNs since existing methods that are based on end-to-end regularized AT with BPTT are not computationally feasible at this scale [8–10, 35]. In Table 2 we present our results obtained by converting robustly pretrained ANNs with layerwise uniform sparsity. Our sparse SNNs yield clean/robust accuracies that proportionally scale with the performance of the baseline ANN, highlighting the importance of the pretrained ANN based sparse weight initialization, e.g., clean/robust acc. under PGD$_{ens}$ with $\epsilon = 2/255$, our 50% sparse SNN: 57.3/26.1 and 90% sparse SNN: 53.7/23.0, while the clean/robust acc. under PGD with 50% sparse ANN: 56.1/37.6 and 90% sparse ANN: 54.4/35.8. Notably, we also show that PGD$_{ens}$ attacks demonstrated a more rigorous evaluation baseline than RGA attacks [45], e.g., our 90% sparse SNN with $\epsilon = 4/255$, RGA: 15.7%, PGD$_{ens}$: 9.2%, and even FGSM$_{ens}$: 14.7%.

Table 2: Evaluations on TinyImageNet with VGG-11 models at different layerwise uniform sparsity levels. We present robust accuracies (%) under white-box attacks with perturbation budgets of $\epsilon = 2/255$ and $\epsilon = 4/255$.

|  |  | Robust ANN | | Robust SNN (**Ours**) | | |
|---|---|---|---|---|---|---|
|  |  | Clean | PGD | Clean | RGA | PGD$_{ens}$ |
|  | Dense | 57.7 | 39.6 / 23.5 | 58.0 | 34.7 / 16.4 | 25.8 / 8.9 |
| Sparse | 50% | 56.1 | 37.6 / 21.9 | 57.3 | 34.7 / 17.2 | 26.1 / 10.1 |
|  | 70% | 55.8 | 37.6 / 21.9 | 56.9 | 34.0 / 17.3 | 25.9 / 10.1 |
|  | 90% | 54.4 | 35.8 / 20.1 | 53.7 | 32.0 / 15.7 | 23.0 / 9.2 |
|  | 95% | 52.5 | 33.7 / 19.4 | 49.4 | 28.2 / 13.3 | 19.6 / 7.2 |

## 5.4. Impact of SNN Sparsity on Energy-Efficiency

Table 3 presents our spiking activity and energy efficiency analyses of VGG-16 models on CIFAR-10. Firstly, we calculate the total number of spikes elicited by each SNN on average for a test sample. Although the spiking activity of models appeared to increase with higher sparsity (see #Spikes column), we also observed that our adversarially robust SNNs generally depicted high *coding efficiency*, individually. Specifically, VGG-16 consists of 276,992 IF neurons operating for $T = 8$ timesteps, which could induce a total of ~2.2M possible spikes for a single inference. However, the dense SNN (top row) was only approx. 12.5% actively spiking ($2.76 \times 10^5 / 2.2 \times 10^6$), or similarly the 99% sparse SNN (bottom row) was only 14.3% actively spiking.

Considering the sparsity structure of each SNN and based on Eq. (12), we then estimate the relative energy efficiency with respect to the dense SNN (see rightmost column in Table 3). We observed that despite the increase in #Spikes, sparsity in fact consistently yielded more

Table 3: Relative energy efficiency estimates of our SNNs with respect to a dense SNN. #Spikes column indicates the total # of spikes elicited on average for a test sample.

| | Sparsity | #Spikes | $\frac{\#\text{Spikes}_{\text{DenseSNN}}}{\#\text{Spikes}_{\text{SparseSNN}}}$ | $\frac{E_{\text{DenseSNN}}}{E_{\text{SparseSNN}}}$ |
|---|---|---|---|---|
| Dense SNN | 0% | $2.76 \times 10^5$ | 1.00 | **1.00** |
| Sparse SNN (Uniform) | 50% | $3.04 \times 10^5$ | 0.91 | **1.50** |
| | 60% | $3.05 \times 10^5$ | 0.90 | **1.84** |
| | 70% | $3.08 \times 10^5$ | 0.89 | **2.42** |
| | 80% | $2.87 \times 10^5$ | 0.96 | **4.13** |
| | 90% | $3.17 \times 10^5$ | 0.87 | **6.12** |
| Sparse SNN (Non-Uniform) | 90% | $3.02 \times 10^5$ | 0.91 | **2.15** |
| | 95% | $3.07 \times 10^5$ | 0.90 | **3.10** |
| | 99% | $3.14 \times 10^5$ | 0.88 | **8.61** |

energy-efficient models. This was due to the decrease in the number of synaptic connections (hence less propagated spikes) largely outweighing the increase in spiking activity (see Appendix B.3 for per-layer spike rates). Overall, we obtained adversarially robust SNNs with layerwise uniform compression rates of up to 10×, which were also up to ~6.1× more energy efficient than a dense and robust SNN. Our SNNs with non-uniform per-layer sparsities achieved up to 100× reduction in the number of parameters, with an estimated 8.6× energy efficiency increase relative to the dense SNN.

# 6. Discussion

We introduce a solution to the problem of achieving sparsity in adversarially robust SNNs, which has not been investigated before. We show that the sparse and robust network connectivity and weights optimized in the ANN domain can be effectively used in a novel hybrid ANN-to-SNN conversion algorithm with post-conversion sparse finetuning. Our approach is superior to end-to-end AT of sparse SNNs in terms of performance and training efficiency, and achieves state-of-the-art robustness against rigorous adversaries. Our energy consumption estimations reveal that the resulting SNNs are also more energy-efficient at high compression rates, than their dense counterparts.

Our robust SNN finetuning method can also be extended with dynamic robust and sparse training methods that enable rewiring of the transferred ANN connectivity [48]. In this work, we focus on unstructured sparsity, since existing robust ANN pruning algorithms exclusively consider these methods. One can also consider structured sparsity in the pretrained ANN, which might be more suitable for hardware oriented memory efficiency gains [49], albeit with limited robustness capabilities [50]. Our work fundamentally differs from recent SNN pruning methods, as these approaches do not consider robustness objectives [51, 52], and we perform robust pruning in the ANN domain.

We evaluate feedforward SNNs on standard machine learning tasks similar to previous SNN robustness studies [8, 9, 35], as opposed to spike-based data streams [53–55]. This enables us to use traditional ANN architectures pretrained on the same task, which is needed for conversion. Through this ANN, we can incorporate existing robust training and pruning schemes from the domain of ANNs to improve the robustness and structure the sparsity of the resulting SNNs. Thus, our contribution is suitable to incorporate existing and future advances in the field of adversarial machine learning with traditional ANNs, towards energy and memory efficient, reliable AI applications.

# Acknowledgments

This research was funded in whole or in part by the Austrian Science Fund (FWF) [10.55776/COE12]. This work was supported by the Graz Center for Machine Learning (GraML), the NSF EFRI grant #2318152, and the "University SAL Labs" initiative of Silicon Austria Labs (SAL) and its Austrian partner universities for applied fundamental research for electronic based systems.

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

# A. Details on the Experimental Setup

## A.1. Datasets and Models

We experimented with CIFAR-10/100 and TinyImageNet datasets. CIFAR datasets both consist of 32×32 dimensional 50,000 training and 10,000 test images, from 10 and 100 classes respectively [56]. TinyImageNet dataset consists of 64×64 dimensional 100,000 training and 10,000 test images from 200 classes [57]. We use conventional data augmentation approaches involving random cropping of images into 32×32 dimensions by padding zeros for at most 4 pixels around it (or random resized crop to 64×64 for TinyImageNet), and randomly performing a horizontal flip. We use VGG-11, VGG-16 [58] and WideResNet [59] architecture with depth 28 and width 4 (i.e., WRN-28-4).

## A.2. Implementation & Training Configurations

Our proposed adversarially robust and sparse ANN-to-SNN conversion methodology is outlined in Algorithm 1. It is an extension of the robust ANN-to-SNN conversion method introduced for densely connected neural network models [11], and the implementations are publicly available at: `https://github.com/IGITUGraz/RobustSNNConversion`.

**Adversarial Pretraining of Dense ANNs:** We perform adversarial training via the TRADES [26] using 10-step PGD with $l_\infty$-norm bounded perturbations of magnitude $\epsilon = 2/255$. We set the TRADES loss trade-off parameter $\lambda_{\text{TRADES}} = 2$ in the main results. We explore stronger AT with the baseline ANN in later ablation experiments with higher $\epsilon$ and $\lambda_{\text{TRADES}} = 6$ in Table B4. We pretrain the ANNs for 100 epochs via momentum SGD, with learning rate of 0.1 and a weight decay of 0.0001.

**Pruning Adversarially Robust ANNs:** For layerwise uniformly and non-uniformly robust pruning, we follow the training configurations from HYDRA [5] and HARP [6], respectively. We perform robust importance score optimization using the same dense ANN training objective for 20 epochs. Based on these learned importance scores, we prune the model for a given $\kappa$. Afterwards, we adversarially finetune sparse ANNs for 30 epochs, using momentum SGD with a learning rate of 0.01 and a weight decay of 0.0001 [5].

**Adversarial Finetuning of Sparse SNNs:** During conversion, we estimate per-layer firing thresholds with the use of 10 training mini-batches of size 64. We generate these inputs via direct coding for $T_c = 100$, and observe the pre-activation values [23]. We choose our initial estimates as the maximum value in the $\rho = 99.7\%$ percentile of the distribution of observed values, similarly to [23, 39]. We use $\lambda = 0.3$ as the scaling factor on these initial estimates, while setting the trainable firing thresholds. Adversarial end-to-end SNN training baselines were implemented with sparse models initialized with $V_{th}^l = 1$ for all layers, similar to [8, 9]. During conversion, we re-use the pretrained ANN batch-norm parameters for the SNN, but now estimate the mean and variance moving average statistics from scratch during robust SNN finetuning phase on the spiking data.

We set the simulation timesteps $T = 8$ and consider non-leaky IF neurons ($\tau = 1$) in all SNNs. We perform robust and sparse SNN finetuning for 80 epochs, using momentum SGD with a initial learning rate of 0.001, a weight decay of 0.0001, and cosine annealing based learning rate schedulers throughout. We use random-step FGSM (RFGSM) [41] based adversarial examples during robust and sparse finetuning of all SNN models, using an $l_\infty$-bounded perturbation budget of $\epsilon = 2/255$, on the regularizer part of our finetuning objective function $\mathcal{L}_{\text{RFGSM}} = D_{\text{KL}}\left(f(\tilde{\boldsymbol{x}}; \mathbf{m} \odot \boldsymbol{\theta}) \| f(\boldsymbol{x}; \mathbf{m} \odot \boldsymbol{\theta})\right)$.

**Computational Overhead:** Since our method also involves an adversarial finetuning stage, it partially suffers from the same computational burden of AT with BPTT. To obtain our models, we perform 100 epochs of dense ANN adversarial training, 20 epochs of robust importance score optimization, and 30 epochs of sparse ANN finetuning. After ANN-to-SNN conversion, we perform adversarial finetuning of the sparse SNN via BPTT for 80 epochs. Given this experimental setting, wall-clock training time comparisons on a single NVIDIA Quadro RTX 8000 GPU are: baseline ANN adversarial training (100 epochs) + robust ANN pruning (50 epochs) + robust SNN finetuning (80 epochs): ~24.5 hours, whereas end-to-end SNN adversarial training for 350 epochs: ~40.6 hours).

It is important to note that SNN-based parameter updates with AT is significantly more time consuming due to the temporal dimension and the need to unroll the network for BPTT also to compute adversarial examples. However, since the majority of our parameter optimization is performed in the ANN domain, our method eventually results in being the more efficient way to achieve sparsity and adversarial robustness with SNNs.

---

**Algorithm 1** Robust and Sparse ANN-to-SNN Conversion

---

1: **Input:** Dataset $\mathcal{D}$, adversarially pretrained sparse ANN parameters $\{\boldsymbol{W}^l, \varphi^l, \omega^l\}_{l=1}^L$, number of calibration samples, calibration sequence length $T_c$, percentile $\rho$, threshold scaling factor $\lambda$, number of finetuning iterations, simulation timesteps $T$, membrane leak factor $\tau$, trade-off parameter $\beta$, attack perturbation strength $\epsilon$.
2: **Output:** Robust and sparse SNN $f$ with parameters $\boldsymbol{\theta'}$ consisting of $\{\boldsymbol{W}^l, \varphi^l, \omega^l\}_{l=1}^L$ and $\{V_{th}^l\}_{l=1}^{L-1}$.
  ▷ Converting adversarially pretrained sparse ANN
3: **Initialize:**  Set weights of SNN $f$ directly from adversarially pretrained ANN parameters $\{\boldsymbol{W}^l, \varphi^l, \omega^l\}_{l=1}^L$. Define spiking LIF neurons for the SNN layers with $\tau$.
4: **for** $l = 1$ **to** $L - 1$ **do**
5:     **for** $c = 1$ **to** #calibration_samples **do**
6:         **for** t = 1 **to** $T_c$ **do**
7:             $\boldsymbol{a}^l \leftarrow$ Store pre-activation values at layer $l$ during forward pass with direct input coding
8:         **end for**
9:         **if** $\max[\rho\text{-percentile of the dist. in } \boldsymbol{a}^l] > V_{th}^l$ **then**
10:             $V_{th}^l = \max[\rho\text{-percentile of the dist. in } \boldsymbol{a}^l]$
11:         **end if**
12:     **end for**
13: **end for**
  ▷ Initialize trainable firing threshold
14: **for** $l = 1$ **to** $L - 1$ **do**
15:     $V_{th}^l \leftarrow \lambda \cdot V_{th}^l$
16: **end for**
  ▷ Initialize binary sparsity mask **M**
17: **for** $l = 1$ **to** $L$ **do**
18:     $\mathbf{M}_{j,i}^l \leftarrow \begin{cases} 1 & \text{if} \quad \mathbf{W}_{j,i}^l \neq 0 \\ 0 & \text{if} \quad \mathbf{W}_{j,i}^l = 0 \end{cases}$
19: **end for**
  ▷ Robust and sparse finetuning of SNN after conversion
20: **for** $i = 1$ **to** #finetuning_iterations **do**
21:     Sample a mini-batch of $(\boldsymbol{x}, y) \sim \mathcal{D}$
22:     $\tilde{\boldsymbol{x}} \leftarrow$ Compute via RFGSM-based inner max. with $\epsilon$
23:     $f(\tilde{\boldsymbol{x}}; \mathbf{m} \odot \boldsymbol{\theta}) \leftarrow$ Compute via direct coded $\tilde{\boldsymbol{x}}$ for $T$
24:     $f(\boldsymbol{x}; \mathbf{m} \odot \boldsymbol{\theta}) \leftarrow$ Compute via direct coded $\boldsymbol{x}$ for $T$
25:     $\mathcal{L}_{\text{reg}}(\boldsymbol{x}, \tilde{\boldsymbol{x}}) \leftarrow D_{\text{KL}}\left(f(\tilde{\boldsymbol{x}}; \mathbf{m} \odot \boldsymbol{\theta}) \| f(\boldsymbol{x}; \mathbf{m} \odot \boldsymbol{\theta})\right)$
26:     $\mathcal{L}_{\text{robust}} \leftarrow \mathcal{L}(f(\boldsymbol{x}; \mathbf{m} \odot \boldsymbol{\theta})) + \beta \cdot \mathcal{L}_{\text{reg}}(\boldsymbol{x}, \tilde{\boldsymbol{x}})$
27:     $\Delta \boldsymbol{W}^l \leftarrow \sum_t \frac{\partial \mathcal{L}_{\text{robust}}}{\partial \boldsymbol{W}^l}$
28:     $\Delta \varphi^l \leftarrow \sum_t \frac{\partial \mathcal{L}_{\text{robust}}}{\partial \varphi^l}$
29:     $\Delta \omega^l \leftarrow \sum_t \frac{\partial \mathcal{L}_{\text{robust}}}{\partial \omega^l}$
30:     $\Delta V_{th}^l \leftarrow \sum_t \frac{\partial \mathcal{L}_{\text{robust}}}{\partial V_{th}^l}$
  ▷ Maintain sparse weight connectivity via masks
31:     **for** $l = 1$ **to** $L$ **do**
32:         $\boldsymbol{W}^l \leftarrow \boldsymbol{W}^l - \eta \left(\Delta \boldsymbol{W}^l \odot \boldsymbol{M}^l\right)$
33:     **end for**
34: **end for**

---

### A.3. Ensemble Adversarial Attacks

We consider adaptive adversaries that implement an ensemble SNN attack strategy as introduced in [11], where the adversary does not only utilize a single input gradient approximation path but via an ensemble of various surrogate gradient options during BPTT.

The ensemble consists of the piecewise linear function [60] with $\gamma_w \in \{0.25, 0.5, 1.0, 2.0, 3.0\}$:

$$\frac{\partial \boldsymbol{o}^l(t)}{\partial \boldsymbol{v}^l(t)} = \frac{1}{\gamma_w^2} \cdot \max\{0, \gamma_w - |\boldsymbol{v}^l(t^-) - V_{th}^l|\}, \tag{13}$$

the exponential surrogate gradient function [61] with $(\gamma_d, \gamma_s) \in \{(0.3, 0.5), (0.3, 1.0), (0.3, 2.0), (1.0, 0.5), (1.0, 1.0), (1.0, 2.0)\}$:

$$\frac{\partial \boldsymbol{o}^l(t)}{\partial \boldsymbol{v}^l(t)} = \gamma_d \cdot \exp\left(-\gamma_s \cdot |\boldsymbol{v}^l(t^-) - V_{th}^l|\right), \tag{14}$$

and the rectangular surrogate gradient function [14] with $\gamma_w \in \{0.25, 0.5, 1.0, 2.0, 4.0\}$:

$$\frac{\partial \boldsymbol{o}^l(t)}{\partial \boldsymbol{v}^l(t)} = \frac{1}{\gamma_w} \cdot \text{sign}\left(|\boldsymbol{v}^l(t^-) - V_{th}^l| < \frac{\gamma_w}{2}\right). \tag{15}$$

We also included the straight-through estimator (STE), where an identity function is used as the surrogate gradient for all spiking neurons during backpropagation [62], backward pass through rate (BPTR), which performs a differentiable approximation by taking the derivative of the spike functions directly from the average firing rate of the neurons between layers [8], and a conversion-based approximation, where spiking neurons are replaced with ReLUs during BPTT [31].

We present a detailed decomposition of the individual components of the ensemble in Table A1. Notably, we observed that altering the shape and parameters of the surrogate gradient can influence the success of the overall adversary on different test samples. Several SNN robustness studies simply consider the only naive adversary case, which only utilizes Eq. (13) with $\gamma_w = 1$, i.e., identical surrogate gradient function to the one used during SNN training.

## B. Further Experimental Results

### B.1. Evaluations of Sparse ANNs and SNNs

We present the numerical evaluation details corresponding to the Figure 2 of the main manuscript, in Tables B1 and B2. These tables also include evaluations of the robust and sparse ANN models used for conversion in obtaining the corresponding robust and sparse SNNs. Note that ANNs are evaluated with standard PGD attacks (10-steps) only for completeness. Our primary goal was to perform comparative analysis of sparse SNN models under ensemble attacks.

Our first observation is that robust ANN pruning with learned importance scores significantly outperforms LWM-based pruning in many aspects of the resulting SNN, particularly in terms of benign performance at high model compression rates. Mainly, we successfully achieve models with very high compression by using layerwise non-uniform sparsity structures, e.g., in Table B1 clean/robust acc. under $\text{PGD}_{\text{ens}}$ with $\epsilon = 8/255$ at 99% SNN sparsity, LWM: 44.9/9.2, Uniform: 69.1/15.3, Non-Uniform: 85.2/22.9.

**Varying the $\text{PGD}_{\text{ens}}$ Number of Iterations:** We investigate the resilience of our SNNs against $\text{PGD}_{\text{ens}}$ attacks with increasing number of iterations, in Table B3. Our main goal is to verify the reliability of gradient approximation with our white-box ensemble attacks. This analysis is commonly performed as a sanity check in robustness evaluations [43, 63], since our main results only included 10-step $\text{PGD}_{\text{ens}}$.

We observe that in all models $\text{PGD}_{\text{ens}}$ tends to get slightly more effective, hence demonstrating reliable behavior without signs of misleading obfuscated gradients confounding our evaluations, e.g., robust accuracy at 30% sparsity with $\text{PGD}_{\text{ens}}$ at $\epsilon = 8/255$, 7-steps: 28.9%, 10-steps: 27.9%, 20-steps: 26.2%, 40-steps: 25.4%.

Table A1: Detailed robust accuracy results for the individual components of an FGSM ensemble attack with $\epsilon = 8/255$ on VGG-16 models with 90% sparsity. Strongest individual attack components in the ensemble against each model are underlined.

| | | End-to-End Adv. Training | | Conversion + |
| | | Random Conn. | ANN Conn. | Sparse FT (**Ours**) |
|---|---|---|---|---|
| | Clean Acc. | 87.4 | 88.2 | **89.5** |
| | FGSM$_{ens}$ | 27.8 | 26.5 | **41.1** |
| Pcw. Linear | $\gamma_w = 1.0$ | 38.79 | 38.30 | **54.81** |
| | $\gamma_w = 2.0$ | 54.49 | 54.44 | **68.01** |
| | $\gamma_w = 3.0$ | 67.60 | 67.13 | **75.65** |
| | $\gamma_w = 0.5$ | 44.18 | 42.51 | **54.87** |
| | $\gamma_w = 0.25$ | 81.28 | 82.12 | **84.04** |
| Exponential | $(\gamma_d, \gamma_s) = (0.3, 0.5)$ | 68.10 | 68.33 | **76.40** |
| | $(\gamma_d, \gamma_s) = (0.3, 1.0)$ | 48.15 | 47.26 | **62.90** |
| | $(\gamma_d, \gamma_s) = (0.3, 2.0)$ | 34.29 | 33.61 | **50.40** |
| | $(\gamma_d, \gamma_s) = (1.0, 0.5)$ | 78.15 | 79.12 | **82.38** |
| | $(\gamma_d, \gamma_s) = (1.0, 1.0)$ | 56.36 | 56.35 | **69.66** |
| | $(\gamma_d, \gamma_s) = (1.0, 2.0)$ | 37.01 | 36.38 | **53.44** |
| Rectangular | $\gamma_w = 0.25$ | 85.47 | 85.96 | **87.82** |
| | $\gamma_w = 0.5$ | 69.23 | 66.97 | **70.84** |
| | $\gamma_w = 1.0$ | 46.51 | 44.56 | **57.32** |
| | $\gamma_w = 2.0$ | 59.25 | 59.52 | **70.93** |
| | $\gamma_w = 4.0$ | 78.66 | 79.60 | **83.28** |
| | Straight-Through Estimation | 85.78 | 86.58 | **88.83** |
| | Backward Pass Through Rate | 42.80 | 41.15 | **57.20** |
| | Conversion-based Approx. | 80.73 | 80.61 | **70.84** |

## B.2. Influence of Adversarial ANN Pretraining on Sparse SNN Robustness

We demonstrate the impact of using heavier adversarial ANN pretraining on the resulting converted SNN, in terms of the transferred robustness gains. In Table B4 we present evaluations of converted SNNs that use different configurations when pretraining the baseline ANNs. The robust and sparse finetuning configuration remains the same with $\beta = 2$, and $\epsilon = 2/255$.

We show that SNNs obtained via conversion are able to leverage and maintain the robustness properties of the baseline ANNs through our robust weight initialization approach. A higher $\epsilon$ during ANN pretraining also proportionally yields SNNs with higher adversarial robustness as seen in Table B4, although we observe a drop in benign accuracy with increasing $\epsilon$ as expected through the robustness-accuracy trade-off. Specifically, we could obtain 90% sparse (uniform) SNNs with clean/robust acc. up to 83.2/40.0, when we perform ANN AT using $\epsilon = 8$ and $\lambda_{\text{TRADES}} = 6$, as opposed to our previous 89.5/26.1.

## B.3. Analysis of Layerwise Spike Rates

In Figure B1, we investigate the per-layer spiking rates of 90% sparse SNNs, to elaborate the emerging differences in energy consumption estimates of the two pruning approaches. Note that in this particular setting, our estimates revealed that layerwise uniform sparsity yields $6.12\times$ more, and non-uniform sparsity yields $2.15\times$ more energy-efficient models than the densely connected SNN.

For layerwise non-uniformly sparse models in Figure B1(b), we can observe higher spike rates in the layers with higher connectivity. Since layers with higher connectivity contain more outgoing connections where spikes are transmitted for AC operations, more energy is consumed overall. Naturally, this effect can occur when we use pruning methods that optimize the connectivity on a per-layer basis. In contrast, models obtained by layerwise uniform pruning in Figure B1(a) show more variant layerwise spike rates. However, this model appears to be less prone to consume significantly more energy in certain layers, since sparse connectivity is uniformly distributed overall.

Table B1: Detailed evaluations of robust and sparse VGG-16 models on CIFAR-10, for different robust ANN pruning approaches and sparsity levels. Robust SNNs are obtained with our conversion method using the corresponding baseline ANNs.

| | Sparsity | Robust ANN | | Robust SNN (Ours) | | |
| | | Clean | PGD | Clean | FGSM$_{ens}$ | PGD$_{ens}$ |
|---|---|---|---|---|---|---|
| Dense NN | 0% | 93.1 | 82.2 / 63.9 / 25.2 | 92.0 | 76.3 / 66.2 / 45.8 | 74.9 / 59.8 / 27.9 |
| Sparse NN (LWM) | 10% | 93.2 | 82.2 / 64.3 / 25.5 | 91.8 | 75.7 / 65.9 / 45.5 | 74.1 / 59.4 / 27.7 |
| | 30% | 93.2 | 82.3 / 63.8 / 25.7 | 92.0 | 76.2 / 65.9 / 45.7 | 74.4 / 59.3 / 28.8 |
| | 50% | 93.2 | 82.1 / 63.6 / 25.3 | 91.8 | 76.3 / 66.0 / 45.9 | 74.8 / 60.0 / 28.4 |
| | 70% | 92.8 | 81.5 / 62.3 / 24.5 | 92.0 | 77.3 / 67.3 / 47.3 | 75.3 / 60.9 / 30.0 |
| | 90% | 91.2 | 77.8 / 56.6 / 20.1 | 88.7 | 69.9 / 59.5 / 40.3 | 68.0 / 54.2 / 26.1 |
| | 95% | 88.7 | 73.4 / 51.1 / 16.1 | 78.7 | 54.2 / 46.1 / 31.9 | 52.6 / 42.0 / 23.0 |
| | 99% | 72.1 | 50.7 / 31.4 / 8.20 | 44.9 | 18.8 / 15.8 / 11.2 | 17.9 / 15.0 / 9.20 |
| Sparse NN (Uniform) | 10% | 93.2 | 82.3 / 64.2 / 25.6 | 91.2 | 75.3 / 64.8 / 43.8 | 73.6 / 59.0 / 28.1 |
| | 30% | 93.3 | 82.3 / 63.9 / 25.5 | 90.8 | 74.6 / 64.4 / 44.8 | 72.5 / 57.9 / 27.9 |
| | 50% | 93.2 | 81.9 / 63.0 / 25.3 | 90.7 | 74.7 / 64.8 / 44.6 | 72.8 / 58.8 / 27.9 |
| | 70% | 93.0 | 81.6 / 62.7 / 24.8 | 90.9 | 74.6 / 64.5 / 43.7 | 72.8 / 58.5 / 28.2 |
| | 90% | 92.3 | 79.8 / 59.5 / 22.2 | 89.5 | 71.6 / 61.0 / 41.1 | 69.9 / 54.7 / 26.1 |
| | 95% | 90.9 | 77.7 / 57.2 / 20.6 | 86.9 | 66.0 / 54.9 / 37.0 | 64.3 / 49.3 / 23.7 |
| | 99% | 84.8 | 68.5 / 47.0 / 14.4 | 69.1 | 40.2 / 33.2 / 23.2 | 38.5 / 29.7 / 15.3 |
| Sparse NN (Non-Uniform) | 50% | 93.8 | 83.2 / 64.9 / 27.0 | 89.0 | 73.3 / 62.2 / 43.0 | 71.4 / 55.8 / 25.1 |
| | 90% | 93.4 | 82.6 / 63.7 / 25.4 | 89.6 | 73.2 / 61.0 / 41.5 | 71.1 / 54.2 / 23.7 |
| | 95% | 93.7 | 82.8 / 63.4 / 25.8 | 88.1 | 69.9 / 58.3 / 38.9 | 68.1 / 52.5 / 23.9 |
| | 99% | 92.2 | 79.8 / 60.3 / 21.8 | 85.2 | 62.5 / 52.9 / 35.0 | 61.0 / 48.5 / 22.9 |

Table B2: Detailed evaluations of robust and sparse WRN-28-4 models on CIFAR-100, for different robust ANN pruning approaches and sparsity levels. Robust SNNs are obtained with our conversion method using the corresponding baseline ANNs.

| | Sparsity | Robust ANN | | Robust SNN (Ours) | | |
| | | Clean | PGD | Clean | FGSM$_{ens}$ | PGD$_{ens}$ |
|---|---|---|---|---|---|---|
| Dense NN | 0% | 67.5 | 46.1 / 27.7 / 8.20 | 69.6 | 43.7 / 30.1 / 14.6 | 40.8 / 23.3 / 5.50 |
| Sparse NN (LWM) | 10% | 67.1 | 46.2 / 28.1 / 8.80 | 69.5 | 44.0 / 30.1 / 14.8 | 40.8 / 23.1 / 5.60 |
| | 30% | 67.3 | 46.2 / 27.9 / 8.60 | 69.2 | 43.8 / 30.0 / 14.9 | 40.7 / 22.8 / 5.70 |
| | 50% | 66.9 | 46.0 / 27.5 / 8.10 | 68.9 | 43.3 / 30.1 / 15.2 | 39.9 / 23.2 / 6.00 |
| | 70% | 66.6 | 44.8 / 26.4 / 7.90 | 67.9 | 41.9 / 28.9 / 14.2 | 38.6 / 21.6 / 5.10 |
| | 90% | 65.3 | 43.5 / 25.6 / 7.50 | 56.6 | 32.5 / 23.2 / 12.3 | 30.0 / 17.6 / 4.30 |
| Sparse NN (Uniform) | 10% | 67.5 | 46.3 / 27.9 / 8.70 | 69.6 | 43.6 / 30.2 / 14.5 | 40.6 / 23.0 / 5.70 |
| | 30% | 66.9 | 46.3 / 27.9 / 8.40 | 69.2 | 43.9 / 30.2 / 15.1 | 40.7 / 23.4 / 6.00 |
| | 50% | 66.8 | 46.0 / 27.5 / 8.10 | 69.1 | 43.6 / 30.2 / 15.0 | 40.5 / 22.8 / 5.90 |
| | 70% | 67.2 | 45.9 / 27.3 / 8.20 | 68.0 | 43.0 / 30.3 / 15.3 | 39.9 / 23.4 / 6.00 |
| | 90% | 66.8 | 46.2 / 27.7 / 8.20 | 63.8 | 38.3 / 27.5 / 14.6 | 35.0 / 20.7 / 5.60 |
| Sparse NN (Non-Uniform) | 50% | 67.1 | 46.0 / 27.3 / 8.20 | 68.9 | 43.0 / 30.4 / 14.8 | 40.2 / 23.1 / 6.20 |
| | 90% | 66.3 | 45.8 / 27.6 / 7.90 | 66.1 | 40.7 / 27.9 / 14.2 | 37.2 / 21.1 / 5.40 |
| | 95% | 66.8 | 46.6 / 28.4 / 7.8 | 62.9 | 36.9 / 25.0 / 12.0 | 33.2 / 18.0 / 3.70 |
| | 99% | 60.7 | 40.5 / 22.9 / 5.70 | 44.3 | 23.7 / 16.9 / 8.10 | 21.5 / 11.5 / 2.30 |

Table B3: Evaluations of robust VGG-16 SNNs on CIFAR-10, with increasing number of ensemble PGD attack iterations.

|  | Clean | PGD$_{\text{ens}}$ (7-steps) | PGD$_{\text{ens}}$ (10-steps) | PGD$_{\text{ens}}$ (20-steps) | PGD$_{\text{ens}}$ (40-steps) |
|---|---|---|---|---|---|
| Dense SNN (0%) | 92.0 | 75.1 / 60.6 / 29.0 | 74.9 / 59.8 / 27.9 | 74.5 / 58.8 / 26.9 | 74.0 / 58.3 / 25.8 |
| Sparse SNN (30%) | 90.8 | 73.1 / 58.5 / 28.9 | 72.5 / 57.9 / 27.9 | 72.3 / 57.1 / 26.2 | 72.3 / 56.7 / 25.4 |
| Sparse SNN (50%) | 90.7 | 73.3 / 59.2 / 28.9 | 72.8 / 58.8 / 27.9 | 72.7 / 58.1 / 26.5 | 72.4 / 57.4 / 25.8 |
| Sparse SNN (70%) | 90.9 | 73.2 / 59.2 / 29.5 | 72.8 / 58.5 / 28.2 | 72.3 / 57.7 / 26.8 | 72.4 / 57.4 / 25.9 |
| Sparse SNN (90%) | 89.5 | 70.3 / 55.5 / 27.2 | 69.9 / 54.7 / 26.1 | 69.3 / 54.4 / 24.8 | 69.4 / 53.8 / 24.3 |

Table B4: Influence of heavier adversarial ANN pretraining, on the resulting converted SNN. We explore the use of a higher TRADES loss regularization strength and larger perturbation $\epsilon$ values, using VGG-16 with 90% sparsity on CIFAR-10.

|  | Adversarial ANN Pretraining with TRADES | | | |
|---|---|---|---|---|
|  | $\lambda_{\text{TRADES}} = 2$ | $\lambda_{\text{TRADES}} = 6$ | | |
|  | $\epsilon = 2/255$ | $\epsilon = 2/255$ | $\epsilon = 4/255$ | $\epsilon = 8/255$ |
| Clean Acc. | 89.5 | 89.6 | 87.2 | 83.2 |
| FGSM$_{\text{ens}}$ | 71.6 / 61.0 / 41.1 | 72.6 / 62.6 / 44.2 | 70.4 / 62.8 / 47.9 | 65.7 / 59.3 / 46.8 |
| PGD$_{\text{ens}}$ | 69.9 / 54.7 / 26.1 | 70.9 / 58.0 / 31.9 | 69.2 / 59.7 / 38.3 | 64.8 / 57.3 / 40.0 |

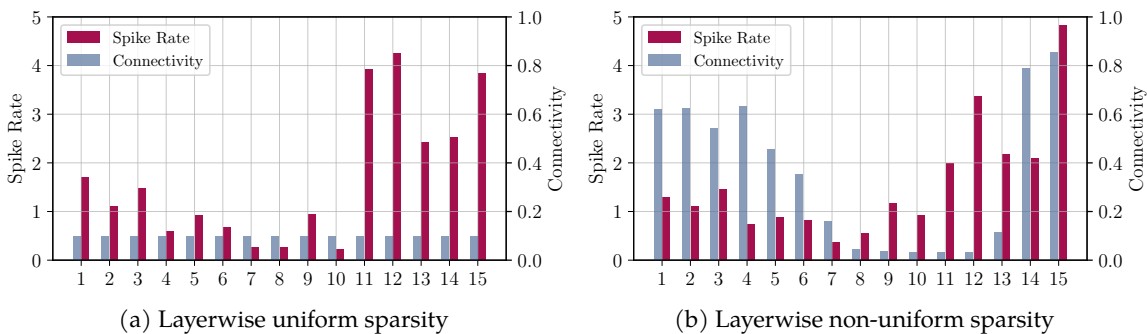

(a) Layerwise uniform sparsity      (b) Layerwise non-uniform sparsity

Figure B1: Comparisons of per-layer average spiking rates of the resulting SNNs with the two pruning approaches. Both models maintain 90% global sparsity (i.e., 10% connectivity) and trained on CIFAR-10 using the VGG-16 architecture.

