# OpenReview forum: "Adversarially Robust Spiking Neural Networks with Sparse Connectivity"
_CPAL.cc/2025/Proceedings_Track — CPAL 2025 (Proceedings Track) Poster_

### Official Review · Reviewer_mHB2 · 2024-12-28
**Novel algorithm but the overall effectiveness is questionable**

**Rating:** 5
**Confidence:** 3

**Review:**

The paper proposes a novel algorithm to convert an adversarially pre-trained ANN to an SNN with sparse connections.  The result is evaluated on a tiny image reporting several orders of compression and energy gain. However, the paper is evaluation is somewhat erroneous because of the following reasons:

The evaluation is using non-structured pruning. However, non-structured pruning does not give the reported benefit of compression since you have to store the places where the connections are eliminated which becomes even more complicated for SNNs, reducing the overall benefit of the proposed method.

---

### Official Review · Reviewer_gRkF · 2025-01-07
**Review of Adversarially Robust Spiking Neural Networks with Sparse Connectivity**

**Rating:** 7
**Confidence:** 2

**Review:**

## Paper Summary

This paper introduces a method to create efficient and robust spiking neural networks (SNNs) by converting them from standard adversarially trained NNs. Instead of directly training sparse and robust SNNs, the authors first train a robust ANN, make it sparse through pruning, and then convert it to an SNN and do some further adversarial fine-tuning. This approach gives networks that are efficient and fast while being robust.

## Strengths

1. The problem of having robust yet energy efficient networks is a practically relevant problem, and spiking neural networks seems like a good way to achieve sparsity in resource-constrained systems. It seems this is the first work to tackle the problem of training robust SNNs, which is an interesting problem.
2. The proposed method leverages pretrained robust NNs, converts to SNN, and then finetunes to get robustness along with sparsity. This is a reasonable formulation and makes sense to leverage the power of pretrained models.
3. The paper is well-structured and the experiments are detailed.
4. The method works well and is able to achieve high sparsity levels with comparable robust performance as the ANN.

## Weaknesses
1. There could be more detail of the proposed methods. For example, I did not understand in Section 3.2, which part of the conversion method is introduced in this paper and which are taken from prior works. What is the threshold-dependent batch norm? What is the difference between this algorithm and the conventional threshold-balancing method [21, 23]. I did not understand what are the training mini-batch forward passes used for firing thresholds? is $rho$ a hyperparameter - it is not introduced anywhere? Does the trainable firing threshold just mean that $\lambda$ is trained?
2. The explanation given for why finetuning is necessary is that we must calibrate the batch-norm layers - what is the notion of calibration here precisely?


## Questions
1. What is the need for further finetuning after converting ANN to SNN? In Figure 1, even without finetuning, the performance is comparable to after finetuning it seems, so how much wall-clock time is added by finetuning? If that finetuning is done with AT with BPTT, wouldn't that be a bottleneck since that is the inefficient part of optimization?
2. In Table 1, why does clean accuracy drop only slightly when comparing conversion + sparse FT vs sparse and robust ANN, but the robust accuracy drops significantly more?
3. If the sparse and robust ANN is also sparse and robust, how does that compare in efficiency vs the SNN?

Overall, I give rating of 7, since my questions are likely because I am not an expert in SNNs. The paper was overall relatively clear and motivated and has comprehensive experiments.

---

### Official Review · Reviewer_NJLD · 2025-01-12
**Review of Submission 14**

**Rating:** 6
**Confidence:** 4

**Review:**

This work propose a framework for obtain a robust spiking neural network, the framework starts from pre-training robust and sparse ANNs and then convert it into spiking NN and recover the performance by adversarial finetuning.

(+) The manuscript is clearly written and easy to follow

(+) The improvements compared with end-to-end training is significant

(+) Experiments convers various models and datasets

(-) Line 34 "Our initial empirical analyses reveal that achieving robustness through standard end-to-end adversarial training with highly sparse feedforward SNNs is computationally challenging and infeasible with spike-based backpropagation through time (BPTT)" Could the author explain more about this with concrete examples, as it's the main motivations of why we need a ANN to SNN framework rather than end-to-end training SNN, and if the computationally is challenging for adversarial training SNN, does this challenge also applys to the adversarial fine-tuning stage of the proposed framework?

(-) How does the model perform against Auto-Attack?

(-) More discussion about why ANN to SNN outperform end-to-end SNN training would further enhance this work. Does the proposed framework involves more training efforts than end-to-end SNN training?

---

### Meta-Review · Area_Chair_FY1W · 2025-02-03

**Recommendation:** Accept (Poster)
**Confidence:** 4

**Metareview:**

This paper presents a novel approach for sparse and adversarial robust conversion of dense ANNs to spiking neural networks (SNNs). The paper is well-written and the main ideas and clearly presented. The authors reported some significant empirical improvements achieving robustness with high levels of sparsity and tested their approach  to various models and datasets. During the rebuttal phase, the authors addressed reviewers' concerns providing additional results on Auto-Attack and provided several clarifications on computational overhead, intuition for improved performance of their approach over end-to-end adversarial training of SNNs, necessity of the adversarial supervised fine-tuning stage they propose, and the use of non-structured pruning at evaluation. Overall, the paper makes valuable contributions to a relevant problem presenting promising results (in terms of energy and memory efficiency) and therefore I recommend accepting it to CPAL.

---

### Decision · Program_Chairs · 2025-02-11

Accept (Poster)